# ETV2 Enhances CXCL5 Secretion from Endothelial Cells, Leading to the Promotion of Vascular Smooth Muscle Cell Migration

**DOI:** 10.3390/ijms24129904

**Published:** 2023-06-08

**Authors:** Ningning Sun, Beyongsam Chu, Dong-Hyun Choi, Leejin Lim, Heesang Song

**Affiliations:** 1Department of Biochemistry and Molecular Biology, Chosun University School of Medicine, Gwangju 61452, Republic of Korea; sunningning0113@daum.net; 2Department of Medical Sciences, Chosun University Graduate School, Gwangju 61452, Republic of Korea; chusam2@naver.com; 3Department of Internal Medicine, Chosun University School of Medicine, Gwangju 61452, Republic of Korea; dhchoi@chosun.ac.kr; 4Advanced Cancer Controlling Research Center, Chosun University, Gwangju 61452, Republic of Korea

**Keywords:** endothelial cells, CXCL5, vascular smooth muscle cells, migration

## Abstract

Abnormal communication between endothelial cells (ECs) and vascular smooth muscle cells (VSMCs) promotes vascular diseases, including atherogenesis. ETS variant transcription factor 2 (ETV2) plays a substantial role in pathological angiogenesis and the reprogramming of ECs; however, the role of ETV2 in the communication between ECs and VSMCs has not been revealed. To investigate the interactive role of ETV2 in the EC to VSMC phenotype, we first showed that treatment with a conditioned medium from ETV2-overexpressed ECs (Ad-ETV2 CM) significantly increased VSMC migration. The cytokine array showed altered levels of several cytokines in Ad-ETV2 CM compared with those in normal CM. We found that C-X-C motif chemokine 5 (CXCL5) promoted VSMC migration using the Boyden chamber and wound healing assays. In addition, an inhibitor of C-X-C motif chemokine receptor 2 (CXCR2) (the receptor for CXCL5) significantly inhibited this process. Gelatin zymography showed that the activities of matrix metalloproteinase (MMP)-2 and MMP-9 increased in the media of VSMCs treated with Ad-ETV2 CM. Western blotting revealed a positive correlation between Akt/p38/c-Jun phosphorylation and CXCL5 concentration. The inhibition of Akt and p38-c-Jun effectively blocked CXCL5-induced VSMC migration. In conclusion, CXCL5 from ECs induced by ETV2 promotes VSMC migration via MMP upregulation and the activation of Akt and p38/c-Jun.

## 1. Introduction

Atherosclerosis is a chronic, progressive inflammatory disease of the arterial vasculature that occurs throughout the body [1,2] and can lead to coronary artery disease, peripheral artery disease, and kidney problems [3,4,5]. Atherosclerosis is the main cause of life-threatening complications such as myocardial infarction and stroke. The biochemical interaction between endothelial cells (ECs) and vascular smooth muscle cells (VSMCs), two types of cells that comprise blood vessels, is fundamental to the development and formation of blood vessels and plays an important role in maintaining the homeostasis of mature vessel tone and arterial remodeling [6]. The vasculature plays a pivotal role in supplying blood and oxygen to all tissues within the body, making it significant in the treatment of conditions involving ischemia and injury-induced regeneration [7]. Endothelial cells have the ability to secrete vasoactive agents and reactive oxygen species (ROS), which can influence the function of vascular smooth muscle cells and subsequently regulate vascular function [8]. There is evidence that the incidence and pathophysiology of atherosclerosis and its accompanying vascular consequences are positively linked to the dysfunction of ECs and VSMCs [1]. In fact, abnormal EC–VSMC interactions may alter the phenotype of VSMCs from contractile to synthetic and the switched VSMCs show a dedifferentiated proliferative or migratory phenotype in pathological or injured vessels [9,10].

The communication of ECs and VSMCs is coordinated by various factors, including physical contact, the extracellular matrix (ECM), and cytokines [1]. For example, direct contact between ECs and VSMCs via junctional molecules such as N-cadherin, connexin 43, intercellular adhesion molecule 1, and vascular cell adhesion molecule 1 contribute to vessel identity and vasculature formation [11,12,13]. In addition, various secreted ECM components from ECs and VSMCs can alter each other’s functions and disrupt their interactions [14,15]. Extracellular vesicles, microRNAs, and cytokines secreted from both cell types are also known to regulate EC–VSMC interactions, leading to the induction of atherogenesis [16,17,18,19]. However, unanswered questions about the coordinating mechanisms of EC–VSMC interactions, as well as unknown factors affecting their interaction leading to atherogenesis remain.

ETS variant transcription factor 2 (ETV2, also known as ER71), a member of ETS family, has been identified as a master regulator of cardiovascular system development and plays a significant role in pathological angiogenesis and EC reprogramming [20]. Although ETV2 is transiently expressed during embryogenesis, several studies have reported its potential function in the pathophysiology of angiogenesis in adults. ETV2 expression increases following vascular injury in ECs and is required for vascular regeneration [21]. In addition, the transduction of ETV2 improves cardiac function and induces vascular regeneration in animal models of myocardial infarction [20].

To directly investigate the role of ETV2 in EC–VSMC interactions, we overexpressed ETV2 in ECs and determined the altered levels of secreted cytokines from ECs. We found that C-X-C motif chemokine 5 (CXCL5) was highly upregulated by ETV2 and that CXCL5 promoted VSMC migration through matrix metalloproteinase (MMP) upregulation via the Akt/p38 signaling pathway. These results provide novel insights into how ETV2 contributes to the EC–VSMC interactions that lead to atherosclerosis.

## 2. Results

### 2.1. Conditioned Medium from ETV2-Transduced Human Umbilical ECs Promotes VSMC Migration

EC–VSMC interactions affect VSMC turnover and regulate the function of VSMCs [1,22]; several reports have confirmed that ETV2 can induce the maturation of ECs, promote vascular development, and be used as a therapeutic agent in myocardial ischemia [20,23,24]. To define the role of ETV2 in EC–VSMC interactions, we constructed adenoviral vectors expressing ETV2 (Ad-ETV2) and observed that ETV2 was successfully expressed in Ad-ETV2-transduced HUVECs using Western blot and immunocytochemistry (Figure 1A,B). We first evaluated the effects of conditioned medium (Ad-ETV2 CM) from ETV2-induced human umbilical ECs (HUVECs) on VSMC migration. As shown in Figure 1C, using the Boyden chamber assay, VSMCs grown in Ad-ETV2 CM exhibited migratory activity that was significantly more increased than that of VSMCs grown in the control CM (Ad-lacZ CM). We observed similar results using a two-dimensional wound healing assay (Figure 1D). We confirmed that Ad-ETV2 CM did not affect cell viability or proliferation (Figure 1E,F). To investigate how Ad-ETV2 CM increased the motility of VSMCs, we determined the altered levels of secreted cytokines in Ad-ETV2 CM from ECs (Figure 1G). CXC chemokines growth-regulated oncogene α/β/γ, chemokine (C-X-C motif) ligand 1 (CXCL1), monocyte chemoattractant protein-1 (MCP-1), monocyte chemotactic protein 2 (MCP-2), and interleukin-6 (IL-6) were found to be increased 2–3 fold in Ad-ETV2 CM compared with in the control CM. Among them, C-X-C motif chemokine 5 (CXCL5) showed the highest increase (a 4.6-fold increase) compared with that in the control CM. In addition, we observed that the mRNA levels of *CXCL1*, *CXCL5*, *MCP-1*, *IL-6*, and *IL-8* were markedly higher in ETV2-transduced HUVECs than in the non-transduced control group (Figure 1H). These results demonstrate that the altered secretion of cytokines from ECs by ETV2 might affect the migratory activity of VSMCs.

### 2.2. CXCL5 Secreted from ECs Facilitates VSMC Migration via the CXC Motif Chemokine Receptor 2 Axis

Since the altered level of CXCL5 was the highest among the various cytokines secreted from ETV2-transduced HUVECs, we investigated the effects of CXCL5 on VSMC migration. In both the Boyden chamber assay (Figure 2A) and wound healing assay (Figure 2B), CXCL5 increased VSMC migration compared with that in the control group and exhibited a similar level compared with that in the tumor necrosis factor (TNF)-α-treated group (TNF-α is known to stimulate VSMC migration). CXCL5 and TNF-α had no significant effect on VSMC viability or proliferation (Figure 2C,D). Since there are reports that CXCL5 binds to C-X-C motif chemokine receptor 2 (CXCR2) to promote angiogenesis, tumor growth, and metastasis [25,26], we examined the role of CXCR2 in VSMC migration. The migration of VSMCs treated with Ad-ETV2-CM was markedly inhibited by a 5 μM concentration of the CXCR2 inhibitor SB225002 within 48 h (Figure 2E). In addition, we found that the treatment with a CXCL5-specific antibody and a CXCR2 inhibitor effectively inhibited the migration of VSMCs induced by CXCL5, as shown in Figure 2F,G, respectively. These results strongly suggest that the enhanced migration of VSMCs induced by the cytokine CXCL5 released from HUVECs by ETV2 is CXCL5 specific and that its mechanism involves the mediation of VSMC migration through its corresponding receptor, CXCR2.

### 2.3. VSMC Migration by CXCL5 Is Mediated by MMP9/MMP13 but Not MMP2

Previous studies have established that VSMCs have two phenotypes: contractile (differentiated, quiescent, and non-migratory) and synthetic (dedifferentiated, proliferative, and migratory) [27,28]. An effective way to identify VSMC phenotypic switching is by examining the expression of specific genes. We used quantitative reverse transcription polymerase chain reaction (RT-qPCR) to examine the effect of CXCL5 on phenotypic markers of VSMCs. Myocardin and α-SMA are associated with the contractile phenotype and KLF4 and CX43 are markers for the synthetic type, all of which are positively related to the migratory activities of VSMCs [29,30]. Our results showed that the mRNA level of CX43 in VSMCs treated with CXCL5 was 3.4-fold higher than that of the control (Figure 3A), whereas the mRNA level of α-SMA was significantly decreased. MMPs are well known to be responsible for VSMC migration [31] and their increased levels are characteristic of the synthetic phenotype of VSMCs (ref). We observed that treatment with CXCL5 increased the mRNA levels of MMP9 and MMP13 but not of MMP2 (Figure 3B). In addition, CXCL5 significantly increased the protein expression of MMP9 and MMP13 in VSMCs (Figure 3C). Zymographic analyses revealed an increase in MMP9 and MMP13, but not MMP2, in the supernatants of VSMCs treated with CXCL5 (Figure 3D,E). These data indicate that CXCL5 induces phenotypic changes in VSMCs and that CXCL5-induced VSMC migration is specifically mediated by MMP9 and MMP13 but not by MMP2.

### 2.4. CXCL5 Stimulates VSMC Migration in an Akt/p38-c-Jun-Dependent Manner

To investigate the specific signaling pathways involved in the CXCL5-driven migration of VSMCs, we selected several key proteins, including Akt, ERK1/2, p38, and c-Jun, to examine their activities. We observed considerable accumulation of phosphorylated Akt and p38 following CXCL5 treatment at the 1 h and 2 h time points. A significant increase in c-Jun phosphorylation was also observed at shorter time points (30 min and 1 h; Figure 4A). The levels of p-Akt, p-p38, and p-c-Jun were significantly elevated by 1.8- to 3-fold. We also demonstrated that Akt, p38, and c-Jun phosphorylation is significantly inhibited by treatment with the CXCR2 inhibitor SB225002 (Figure 4B). We also investigated changes in the migratory properties of VSMCs after treatment with several phosphorylation inhibitors using a wound healing assay. The results demonstrated that the inhibitors of Akt, ERK1/2, p38, and c-Jun phosphorylation inhibited VSMC migration to varying degrees (Figure 5A). Compared with other inhibitors, U0126 (an ERK1/2 inhibitor) did not regulate CXCL5-induced VSMC migration well. The expression of MMP9 and MMP13 at the cellular level induced by CXCL5 was significantly inhibited by A6730 (an Akt inhibitor), SB203589 (a p38 inhibitor), and SP600125 (a c-Jun inhibitor). However, U0126 pretreatment did not alter the intracellular expression of MMP13 (Figure 5B). Furthermore, gelatin and collagen zymography analyses indicated that phosphorylation inhibitors decreased the activities of MMP9 and MMP13 released by VSMCs (Figure 5C). ERK1/2 inhibitors, however, had a contradictory role in inhibiting MMP9 and MMP13 secretion and activities following quantification. Only the secretion of MMP13 was not affected by the ERK1/2 inhibitor, as shown in Figure 5B. The complexity of the ERK1/2 signaling pathway has been demonstrated. The regulatory mechanism governing the synergy or antagonism between cellular signaling pathways is extremely complex and remains unknown.

To further investigate the role of Akt, ERK1/2, p38, and c-Jun in the CXCL5/CXCR2 axis promoting VSMC migration, the cells were treated with Akt, ERK1/2, p38, and c-Jun phosphorylation inhibitors. As shown in Figure 5C, Akt phosphorylation inhibitors decreased the phosphorylation of proteins other than ERK1/2, indicating that Akt is upstream of this signaling pathway. In addition, p38 phosphorylation inhibitors dramatically reduced the levels of Akt and its own phosphorylation. However, this treatment did not affect c-Jun phosphorylation. In contrast to A6730, the p38 phosphorylation inhibitor SB203580 did not reduce Akt phosphorylation to a comparable level. In conclusion, these findings show that CXCL5 induces VSMC migration via an Akt/p38-c-Jun-dependent pathway.

## 3. Discussion

Several studies have shown that abnormal EC–VSMC interactions promote atherosclerosis development. The dysfunction of ECs induces decreased bioavailability or abnormal activity of vasoactive substances and then induces the recruitment of immune cells to the vessel wall, whereas altered VSMC function promotes abnormal migration and proliferation, leading to the dysregulation of vascular tone and atherosclerosis [32,33]. ETV2 has been reported as a potential therapeutic target for myocardial infarction [34] and a key regulator of vascular regeneration in hind limb animal models [20] through endothelial remodeling. However, the effects of ECs reprogrammed by ETV2 on EC–VSMC interactions remain unclear.

Here, we observed that the secreted CXCL5 from ECs induced with ETV2 promotes VSMC migration via the CXCR2 axis and the Akt/p38-c-Jun signaling pathway. Furthermore, we showed that the increased migration of VSMCs induced by CXCL5 was due to the upregulation of MMP9 and MMP13. This finding suggests a novel mechanism by which the modulation of ECs by ETV2 increases the migratory nature of VSMCs. When ETV2 was tranduced in HUVECs, we observed that the secreted levels of several cytokines, including CXCL1, CXCL5, IL-8, and MCP-2, were altered in the culture media. The results showed that ECs secreted a large number of cytokines, among which the protein secretion and mRNA expression of CXCL5 were higher than those of other cytokines. Indeed, many oncological studies have shown that biological responses, including immune reactions and phenotypic changes in cells and tissues, are organized by cytokines and chemokines, especially CXCL5. However, few studies have investigated this in cardiovascular disease. Our results demonstrate that ETV2 affects the expression levels of cytokines, including CXCL5, and may alter the phenotypic behavior of VSMCs. Therefore, we selected CXCL5 and investigated the mechanism by which CXCL5 affects the migratory properties of VSMCs.

Next, we observed that CXCL5 had a similar ability to promote VSMC migration as TNF-α, a known cytokine that promotes VSMC migration. The blocking experiment using CXCL5 antibodies showed that the increased migration of VSMCs was CXCL5 specific (Figure 2F). Results showing that treatment with a CXCR2 inhibitor significantly inhibited VSMC migration in ETV2-induced CM or CXCL5 (Figure 2E,G) were consistent with those of a previous report, wherein CXCL5 promoted cell migration by interacting with the cell surface chemokine receptor CXCR2 in tumors [35]. These results are supported by a previous finding that CXCL5 overexpression and exogenous administration contribute to the proliferation and migration of HeLa cells, effects that are mediated by CXCR2 [36]. Therefore, our findings show that the production of CXCL5 in ECs in response to ETV2 stimulation is a critical part of VSMC migration that is mediated by CXCR2.

To further characterize the molecular mechanism by which CXCL5 in promotes VSMC migration, we focused on Akt- and ERK1/2-dependent signaling pathways. Indeed, a number of studies have demonstrated that signaling pathways such as Akt/p38, ERK1/2/p-38, Akt/NF-κB, and PI3K/Akt are involved in the migration process of various cell types [37,38,39,40]. In our study, the phosphorylation of Akt, p38, and c-Jun was significantly increased following CXCL5 treatment. We examined the phosphorylation of ERK1/2 and found that it was not altered by CXCL5 treatment. These results are partially inconsistent with those of previous studies. The CXCL5/CXCL2 axis also promotes bladder cancer cell migration by upregulating MMP2/MMP9, which is an Akt-dependent pathway [39]. Upregulated CXCL5 in patients with cervical cancer increases the oncogenic potential of HeLa cervical cancer cells by increasing the phosphorylation of ERK1/2 and Akt and the gene expression of CXCR2 [36]. Considering the results of other studies on CXCL5 molecular signal transduction, ERK1/2-signaling-induced cell migration is mainly generated in tumor cells, which may be due to the activation of ERK1/2 by multiple growth factors and their receptors in tumor cells. In conclusion, VSMC migration induced by CXCL5 is dependent on the Akt signaling pathway but not on ERK1/2 signaling.

The stability of atherosclerotic plaques is significantly influenced by the activities of MMPs and matrix-degrading metalloproteinases, owing to their role in controlling the migration, growth, and survival of VSMCs [41]. Among the MMPs, MMP2 and MMP9 are thought to be involved in VSMC migration [42]. Many reports have shown that MMP expression is affected by various factors, including integrins, integrin-linked kinases, and cytokines [43]. Several cytokines such as MCP2, CXCL1, and IL-8 promote migration via activation of the ERK/MMP2/9 signaling axis [16]. Accumulating data have shown that the P38, Akt, and ERK signaling pathways promote migration by upregulating MMP2 or MMP9 in VSMCs [9,10]. In the present study, we used zymographic analyses to find that CXCL5 significantly upregulated the activities of MMP9 and MMP13 but did not alter the activity of MMP2. Furthermore, pretreatment with an Akt inhibitor (A6730), a p38 inhibitor (SB203580), and a c-Jun inhibitor (SP600125) directly inhibited the activities of MMP9 and MMP13. However, ERK1/2 inhibitors inhibited VSMC migration and MMP9 and MMP13 activities with marginal effects. Despite previous reports that the ERK signaling pathway is involved in the activation of MMP2 in cancer cells [44] and that MMP2 secretion from human ciliary muscle cells is a PKC- and ERK1/2-dependent process [45], we conclude that CXCL5 promotes the migration of VSMCs in an ERK1/2-independent pathway and that the Akt/p38 signaling pathway governs the migration of VSMCs induced by CXCL5. In addition, ETV2 has been reported as a potential therapeutic target for myocardial infarction and a key regulator of vascular regeneration in hind limb animal models through endothelial remodeling. Following injury, the expression of ETV2 was readily detected in endothelial cells. Subsequently, there was a notable increase in vessel formation and neovascularization induced by Etv2 [21,46]. Because vascular injury plays a crucial role as an initial trigger in the development of different vascular disorders, such as atherosclerosis [47], ETV2 may serve as an important factor linking endothelial cells and smooth muscle cells in response to vascular injury.

In conclusion, the present study indicates that the level of CXCL5 is increased in HUVECs reprogrammed with ETV2. Through the activation of the Akt/p38-c-Jun signaling pathway, CXCL5 induces the expression of MMP9 and MMP13, leading to the promotion of VSMC migration (outlined in Figure 5D). These results greatly increase our understanding of vascular remodeling by ETV2 and the molecular mechanisms by which ECs reprogrammed by ETV2 promote VSMC migration in cardiovascular disease.

## 4. Materials and Methods

### 4.1. Reagents

Recombinant human CXCL5 and TNF-α were purchased from Prospecbio (Ness-Ziona, Israel). Specific Akt, ERK1/2, p38, and c-Jun pathway inhibitors (A6730, U0126, SB203580, and SP 600125, respectively) were purchased from Sigma/Aldrich Chemical Co. (St. Louis, MO, USA). Antibodies to MMP2, MMP9, MMP13, p-Akt, Akt, p-ERK1/2, ERK1/2, p-p38, p38, p-c-Jun, and c-Jun was purchased from Cell Signaling Technology (Danvers, MA, USA). Anti-rabbit or anti-mouse HRP-conjugated secondary antibodies were bought from Santa Cruz Biotechnology (Santa Cruz, CA, USA).

### 4.2. Cell Cultures

Human umbilical vein endothelial cells (HUVECs) were purchased from PromoCell (PromoCell, C-12200). HUVECs were cultured in an endothelial cell growth medium kit (PromoCell, C-22110) with 1% penicillin/streptomycin (Capricorn Scientific, PS-B; Ebsdorfergrund, Germany). Vascular smooth muscle cells (VSMCs) were isolated from the thoracic aorta of a 7-week-old male Sprague-Dawley rat using previously published procedures [48]. Briefly, the rat was sacrificed with an overdose injection of Zoletil 50 and Rompun. Then, the rat was disinfected with povidone. The chest was opened and the thoracic aorta was dissected using micro-dissecting scissors to open the aorta and remove the endothelial layers by gentle scraping. The aorta was cut until 1–2 mm^3^ and treated with collagenase and elastase (Sigma, #E1250) for 2 h at 37 °C. The cells were transferred to a tube containing the cell culture medium and centrifuged. The cell pellets were washed with PBS 3 times. Finally, VSMC pellets were resuspended in the cell culture medium and cultured in a cell incubator. VSMCs were cultured in DMEM (Capricorn Scientific, DMEM-HPA; Ebsdorfergrund, Germany) with 10% FBS (Capricorn Scientific, FBS-11A; Ebsdorfergrund, Germany), 5% SMGS (Gibco, S-007-25), and 1% penicillin/streptomycin. All cells were cultured at 37 °C in a humidified environment containing 5% CO_2_.

### 4.3. Construction of Adenoviral Vector

A recombinant adenovirus (named Ad-ETV2) expressing full-length human ETV2 and a C-terminal flag tag was constructed using the ViraPowerTM Adenovirus Expression System (Invitrogen, Waltham, MA, USA). An adenoviral vector expressing lac-Z-β-galactosidase (named Ad-lacZ) was used as infection control. These constructs were then subjected to LR recombination reactions with the pAd/CMV/V5/DEST gateway vector to generate adenoviral expression clones. The constructed adenoviral vectors were transfected into HEK 293A cells after digestion with Pac I restriction enzyme using lipofectamine 2000 reagent (Invitrogen life technologies). The adenovirus particles were titrated using the Adeno-XTM qPCR titration kit (Takara, Clontech).

### 4.4. Cytokine Antibody Array

HUVECs were seeded at 3.3 × 10^5^ cells/well in 6-well cell culture plates, starved with serum medium (EC serum-free medium: EC medium, 8:2) for 12 h, and then infected with ETV2 virus for 72 h. The supernatant of the cell culture was collected. Cytokines in the supernatant were analyzed using the Ray Bio^®^ C-Series Human Cytokine Antibody Array C3 (Ray Bio, AAH-CYT-3-4). Data are shown as the mean ± SD of at least three independent experiments.

### 4.5. RNA Isolation and Quantitative RT-qPCR

The expression levels of various genes were analyzed by an RT-qPCR assay. Preparations of total RNA and cDNA synthesis from cells were performed as described previously [49]. The cells were seeded into a 6-well plate at a density of 5 × 10^5^ cells/mL and cultured for 24 h. The cells were treated with negative control (DMSO), CXCL5, and TNF-α for 6 h. The total RNA was extracted with TRIzol Lysis Reagent (QIAGEN) according to the instructions provided by the manufacturer. The total RNA concentration of each sample was measured at 260 nm using a spectrophotometer. The total RNA was synthesized using the Prime Script™ 1st-Strand cDNA Synthesis Kit (Takara) for reverse transcription. Real-time quantitative PCR was performing using the SYBR Green method using Applied Rotor-Gene 3000™. Gene expression was normalized to GAPDH. Relative mRNA expression levels were analyzed quantitatively using the ΔΔCt method using Rotor-Gene 6 software (Corbett-research). All primer sequences used for qPCR are shown in Table 1.

### 4.6. MTT Cell Viability Assay

For the 3-(4,5-dimethylthiazol-2-yl)-2,5-diphenyltetrazole (MTT) assay, VSMCs were treated with Ad-lacZ CM or Ad-ETV2 CM. After exposure, cells were washed with phosphate-buffered saline (PBS) and then MTT (Sigma Aldrich, St. Louis, MI, USA) solution (pre-pared in Opti-MEM 10% FBS; concentration of 0.3 mg/mL) was added to the cells and incubated for 3 h. Data are expressed as the mean ± SD of at least three independent experiments.

### 4.7. BrdU Proliferation Assay

VSMCs cultured with Ad-lacZ CM or Ad-ETV2 CM were incubated with BrdU (10 µM) for 3 h and stained for BrdU according to the manufacturer’s instructions (APC BrdU Flow Kit, BD Pharmingen). Data are presented as the mean ± SD of at least three separate experiments.

### 4.8. Cell Migration Assay

VSMC migration was measured using the Boyden chamber assay and wound healing assay. For the Boyden chamber assay, gelatin coating was performed on polycarbonate filters in the upper chamber. After 1 h, the number of cells in DMEM was measured and 3 × 10^4^ cells/400 μL inside the insert of the upper chamber and 600 μL of DMEM containing 10% FBS was added to each well in the low chamber, followed by incubations for 6 h and 12 h. Cells attached to the submembrane surface were stained with hematoxylin and eosin at room temperature and counted using a light microscope. For the wound healing assay, VSMCs were seeded at a concentration of 5 × 10^5^ cells/well in 6-well plates. Next, each well was passed over a cell monolayer to create an artificial wound. At 24 and 48 h post-wounding, images of the scratched wounds were taken with a light microscope. The wound width at time zero was subtracted from the wound width to obtain net wound closure. Data are shown as the mean ± SD of a minimum of three independent experiments.

### 4.9. Gelatin and Collagen Zymography

VSMCs (2 × 10^5^ cells/well) were seeded in 6-well plates and cultured for 24 h. The culture medium was obtained. Samples were separated with 10% SDS–polyacrylamide gel with 0.8% gelatin (for MMP2 and MMP9 detection) and 0.3 mg/mL collagen (for MMP13 detection) using the following conditions: 120 V for 5 h. After electrophoresis, the gels were washed with 2.5% Triton-X-100 and incubated with incubation buffer for 24 h at 37 °C and then stained with Coomassie blue. Quantification was performed using Image J.

### 4.10. Western Blot Analysis

VSMCs were lysed in chilled RIPA buffer (50 mM Tris, 150 mM NaCl, pH 8.0, 0.5% deoxycholate, 1% NP-40, 0.1% SDS, protease inhibitor cocktail, and 1 mM phenylmethylsulfonyl fluoride (PMSF)) for 1 h at 4 °C. Cell lysates were clarified by microcentrifugation (14,000 rpm) in a microcentrifuge at 4 °C. Protein concentrations were determined using the Bradford Protein Assay Kit. Equal amounts of cell lysate samples were separated by SDS-PAGE. Next, protein bands were transferred to polyvinylidene difluoride (PVDF, Bio-Rad) membranes. The membranes were incubated with the primary antibody overnight, followed by incubation with the HRP-conjugated secondary antibody for 1 h at room temperature. Immunoreactive proteins were detected with an enhanced chemiluminescence (Roth, #3) system as described previously [43].

### 4.11. Statistical Analysis

Data are expressed as mean ± SD. Student’s *t*-test was used to analyze the differences between groups. Statistical comparisons among multiple groups were performed using analysis of variance [3]. Results were considered statistically significant when the *p*-value was less than 0.05. All the statistical analyses were performed using Prism (GraphPad Software; Boston, MA, USA).

## Figures and Tables

**Figure 1 ijms-24-09904-f001:**
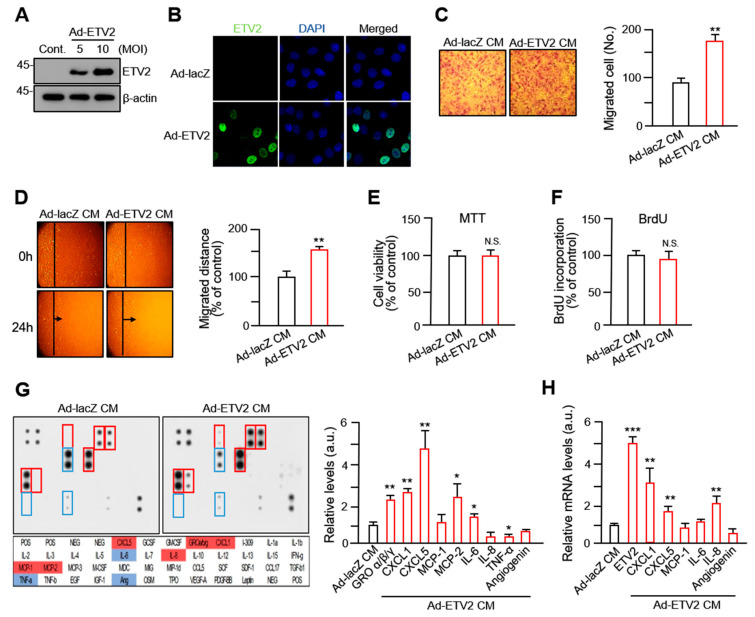
Conditioned medium from ETS variant transcription factor 2 (ETV2)-transduced endothelial cells (ECs) promotes vascular smooth muscle cell (VSMC) migration. (**A**,**B**) Expression levels of ETV2 in HUVECs were evaluated using Western blot and immunocytochemistry, respectively. (**C**) Migration of VSMCs detected by the Boyden chamber assay. Scale bars, 100 μm. Three different areas of migrated cells were counted for each data point. All values are presented as mean ± standard deviation (SD). ** *p* < 0.01 versus Ad-lacZ CM group. (**D**) VSMC migration in wound healing assays. Cell monolayers were wounded (black lines) and treated with Ad-lacZ CM or Ad-ETV2 CM. After 48 h, the migrated cells were photographed and quantified (bar graph). ** *p* < 0.01 versus Ad-lzcZ CM group. (**E**) Cell viability MTT assay and (**F**) BrdU proliferation assay of VSMCs treated with different CMs for 24 h. All values represent mean ± SD. N.S.: not significant. (**G**) Graphical representation of cytokine expression. Cytokine array blots were probed with Ad-lacZ CM and Ad-ETV2 CM for 48 h post-treatment. The blots marked in red are the cytokines that were upregulated between the paired samples. The blots marked in blue were downregulated. The alignment of cytokines in duplicates on the cytokine array is shown in the left lower panel. In the cytokine array, protein expression was evaluated by gray densitometric analysis. The mean value of pixel densities in controls was set as 1.0 and the fold change was calculated for each protein. All values are presented as mean fold change ± SD. * *p* < 0.05 and ** *p* < 0.01 versus Ad-lacZ CM group. Data are representative of three individual experiments (*n* = 3). (**H**) The relative mRNA expression of targeted genes was measured by quantitative reverse transcription polymerase chain reaction (RT-qPCR) in human umbilical ECs (HUVECs) with a control or ETV2 adenovirus. The mean value of the mRNA level in controls was set as 1.0 and the fold change was calculated for each mRNA. All values represent mean ± SD. ** *p* < 0.01 and *** *p* < 0.001 versus Ad-lacZ CM group. All images shown are representative of those obtained from at least three independent experiments.

**Figure 2 ijms-24-09904-f002:**
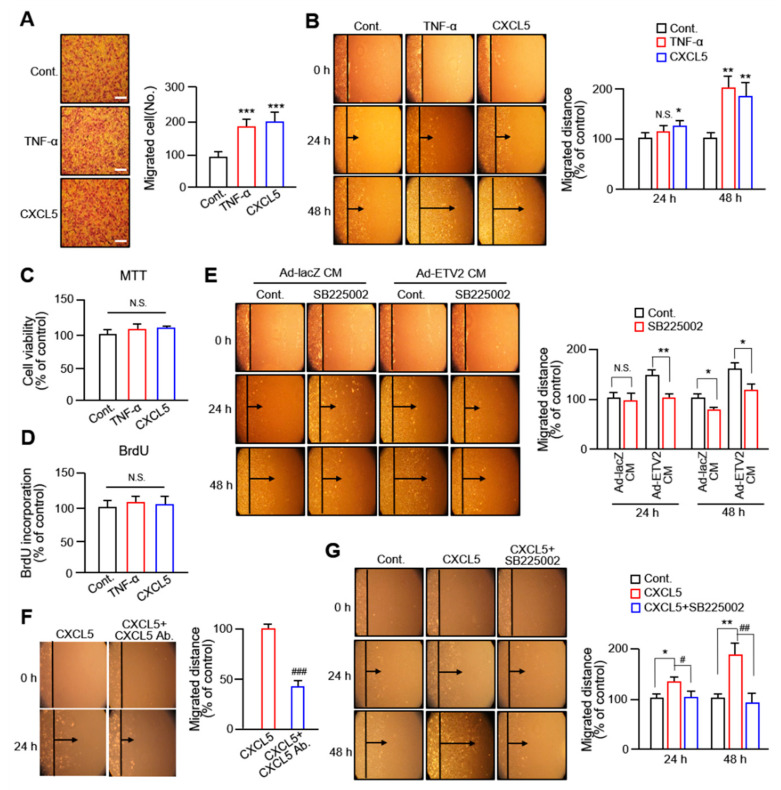
CXC motif chemokine 5 (CXCL5) facilitates VSMC migration via the CXCL5/CXC motif chemokine receptor 2 (CXCR2) axis. (**A**) Representative images of the Boyden chamber assay of VSMCs treated with control, 10 ng/mL tumor necrosis factor (TNF)-α, or 100 ng/mL CXCL5. Scale bars, 100 μm. All values are presented as mean ± standard deviation (SD). *** *p* < 0.001 versus control group. (**B**) VSMC migration was assessed by wound healing assay after 24 h and 48 h of treatment with a designated chemical; untreated cells were used as controls. All values are presented as mean ± standard deviation (SD). * *p* < 0.05 and ** *p* < 0.01 versus control group. (**C**) MTT assay and (**D**) BrdU proliferation assay of VSMCs treated with the control, TNF-α, or CXCL5 for 24 h. N.S.: not significant. (**E**) Representative images of wound healing assay in CM-treated VSMCs w/ or w/o SB225002. All values are presented as mean ± standard deviation (SD). * *p* < 0.05 and ** *p* < 0.01 versus non-treated control. (**F**) Representative images of the blocking experiment for CXCL5 using its specific antibody in VSMCs. All values are presented as mean ± standard deviation (SD). ^###^ *p* < 0.001 of CXCL5 with its antibody versus CXCL5 alone. (**G**) Representative phase-contrast microscope images showing the area covered by the cells at 0, 24, and 48 h after wounding. Percentage of migratory distance change was determined by the rate of cells moving towards the scratched area upon a certain time. All values represent mean ± standard deviation (SD). * *p* < 0.05 and ** *p* < 0.01 versus non-treated control; ^#^ *p* < 0.05 and ^##^ *p* < 0.01 versus CXCL5.

**Figure 3 ijms-24-09904-f003:**
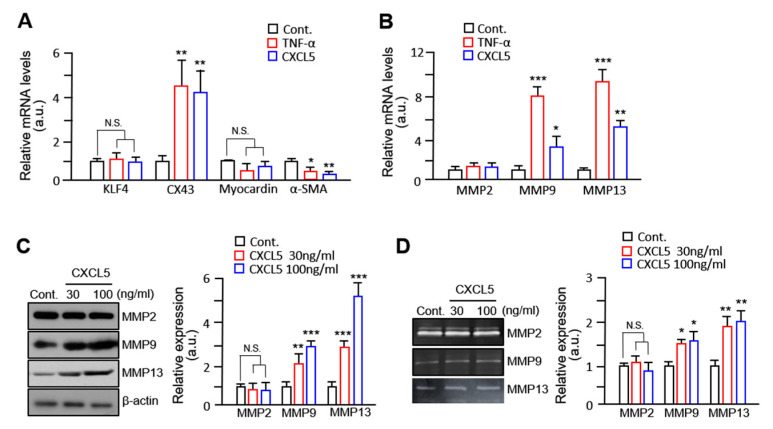
CXCL5 is involved in the increase in matrix metalloproteinase (MMP)9/MMP13. (**A**,**B**) mRNA expression of KLF4, CX43, MMP9, MMP13, myocardin, and α-SMA in control, TNF-α-induced or CXCL5-induced groups was detected by RT-qPCR, and glyceraldehyde 3-phosphate dehydrogenase (GAPDH) was used as a normalization control. (**C**) Cells were treated with 30 ng/mL CXCL5 and 100 ng/mL CXCL5 and lysed; their lysates were immunoblotted with MMP2, MMP9, MMP13, and β-actin antibodies. (**D**) The activities of MMP2, MMP9, and MMP13 in the media from cells treated with CXCL5 were evaluated by gelatin zymography or collagen zymography. All values are presented as mean ± SD. * *p* < 0.05, ** *p* < 0.01, and *** *p* < 0.001 versus control. N.S.: not significant.

**Figure 4 ijms-24-09904-f004:**
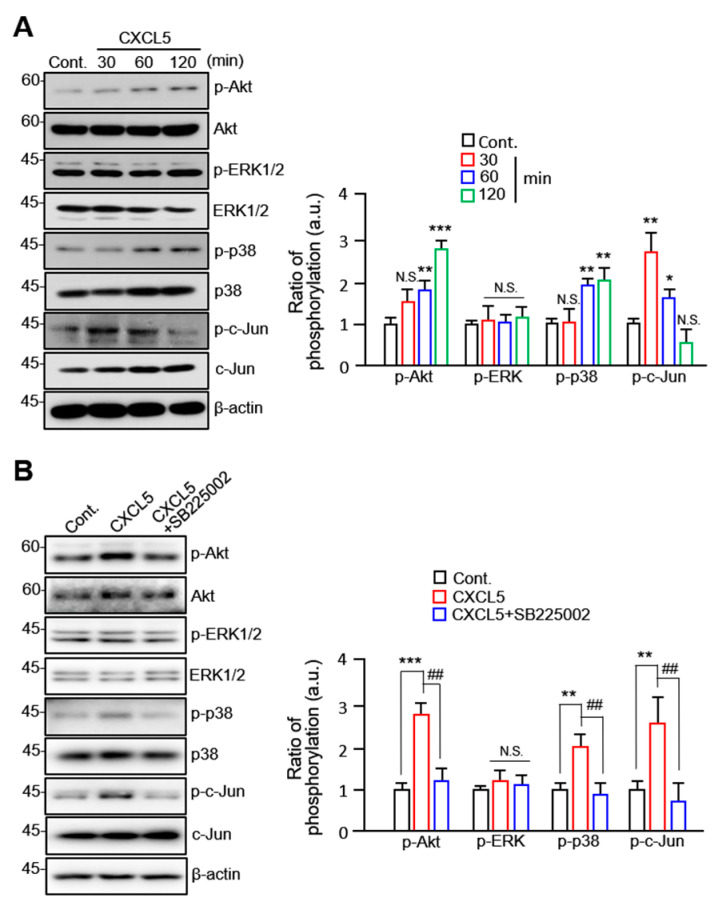
CXCL5 activates the signaling pathway via CXCR2. (**A**) Cells were treated with 30 ng/mL CXCL5 for the indicated periods. Representative Western blots of p-Akt, Akt, p-ERK, ERK, p-p38, p38, p-c-Jun, and c-Jun in VSMCs are shown in the left panel. The right panel is the quantification of phosphorylated p-Akt, p-ERK, p-p38, and p-c-Jun normalized to total protein. All values are presented as mean ± SD. * *p* < 0.05, ** *p* < 0.01, and *** *p* < 0.001 versus control; N.S.: not significant. (**B**) VSMCs were treated with 100 ng/mL CXCL5 for 2 h or pretreated with 5 μM SB225002. p-Akt, p-ERK, p-p38, and p-c-Jun were assessed by Western blotting, followed by quantification and normalization by ImageJ. The right panel represents quantitative data of relative expression. All values are presented as mean ± standard deviation (SD). ** *p* < 0.01 and *** *p* < 0.001 versus non-treated control; ^##^ *p* < 0.01 versus CXCL5.

**Figure 5 ijms-24-09904-f005:**
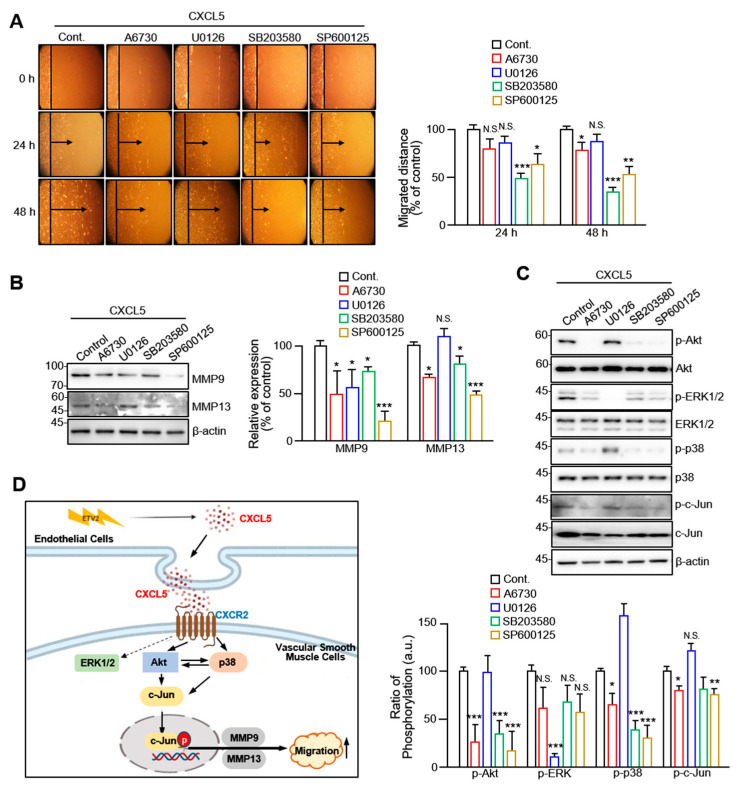
CXCL5 stimulates VSMC migration in an Akt-, p38-, and c-Jun-dependent manner. (**A**) VSMC migration was assessed by a two-dimensional (2D) wound healing assay after 24 h and 48 h. Representative images from the migration assay and statistical results using VSMCs pretreated with 10 μM A6730 (an Akt inhibitor), 10 μM U0126 (an ERK inhibitor), 20 μM SB203580 (a P38 inhibitor), and 10 μM SP600125 (a c-Jun inhibitor). The right panel shows quantitative data. (**B**) Cells were pretreated with 10 μM A6730, 10 μM U0126, 20 μM SB203580, and 10 μM SP600125, then treated with 100 ng/mL CXCL5, lysed, and immunoblotted with MMP9, MMP13, and β-actin antibodies. (**C**) VSMCs were incubated with 10 μM A6730, 10 μM U0126, 20 μM SB203580, and 10 μM SP600125 for 30 min. All groups were treated with 100 ng/mL CXCL5 for 2 h. Cell lysates were collected for a Western blot analysis. p-Akt, p-ERK, pP38, and p-c-Jun were quantified and normalized by ImageJ. The lower panel shows quantitative data. (**D**) Schematic of the CXCL5-Akt/p38-c-Jun signaling model during the migration of VSMCs. All values are presented as mean ± SD. * *p* < 0.05, ** *p* < 0.01, and *** *p* < 0.001 versus control group. N.S.: not significant.

**Table 1 ijms-24-09904-t001:** The sequences of primers used for RT-q-PCR.

Gene	Forward	Reverse
*ETV2*	ATTCAGCTGTGGCAGTTCCT	CCGAAGCGGTACGTGTACTT
*CXCL1*	GCGCCCAAACCGAAGTCATA	ATGGGGGATGCAGGATTGAG
*CXCL5*	CGGGAAGGAAATTTGTCTTGA	AGCTTAAGCGGCAAACATAGG
*IL-6*	GCACTGGCAGAAAACAACCT	TCAAACTCCAAAAGACCAGTGA
*MCP-1*	TTCCCCTAGCTTTCCCCAGA	TCCCAGGGGTAGAACTGTGG
*Angiogenin*	TTCCTGACCCAGCACTATGATG	CGTCTCCTCATGATGCTTTCAC
*Il-8*	CTCTTGGCAGCCTTCCTGATT	ACTCTCAATCACTCTCAGTTCT
*KLF4*	TCAAGAGCTCATGCCACCGG	CTCGCCTGTGTGAGTTCGCA
*CX43*	GGCCTTCCTGCTCATCCA	GGGATCTCTCTTGCAGGTGTAGA
*Myocardin*	CAGAAAGTGACAAGAACGATACAG	TGAAGCAGCCGAGCATAGG
*α-SMA*	AACTGGTATTGTGCTGGACTCTGG	CACGGACGATCTCACGCTCAG
*MMP2*	GCGATGGCAAGGTGTGGTGT	GTACCAGTGTCAGTATCAGC
*MMP9*	AGGCGCCGTGGTCCCCACTTACTT	GCAGGGTTTGCCGTCTCCGTTGCC
*MMP13*	TCGCATTGTGAGAGTCATGCCAACA	TGTGGTTCCAGCCACGCATAGTCA

## Data Availability

The data presented in this study are available on request from the corresponding author. The data are not publicly available due to privacy.

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
