# Peer review of "ETV2 Enhances CXCL5 Secretion from Endothelial Cells, Leading to the Promotion of Vascular Smooth Muscle Cell Migration"

_ijms, 2023, doi:10.3390/ijms24129904_

Round 1
Reviewer 1 Report
The authors lack 1 key experiment that could strengthen their Results and support their conclusions. Use of the CXCR2 or CXCL5 inhibitor in the presence/absence of ETV2 conditioned media on VSMC migration and downstream Akt activation. Otherwise the specificity of ETV2 up regulating CXCL5, as opposed to another factor, is in question. Similarly, to see if these effects are dependent on ETV2, a knock-out cell line should be developed (either using siRNA KO or a stable CRISPR-Cas9 mediated KO clone).
No major English language errors noted. Simple proofreading should suffice.
Author Response
Reviewer #1:
We thank the reviewer for providing the helpful suggestions to improve our manuscript. Our responses to the reviewer’s suggestions/comments are detailed below:
- The authors lack 1 key experiment that could strengthen their Results and support their conclusions. Use of the CXCR2 or CXCL5 inhibitor in the presence/absence of ETV2 conditioned media on VSMC migration and downstream Akt activation. Otherwise the specificity of ETV2 up regulating CXCL5, as opposed to another factor, is in question. Similarly, to see if these effects are dependent on ETV2, a knock-out cell line should be developed (either using siRNA KO or a stable CRISPR-Cas9 mediated KO clone).
We would like to emphasize that we demonstrated the inhibitory effect of the CXCL5-specific antibody or the CXCR2 inhibitor, SB225002, on the enhanced migration of VSMCs induced by Ad-ETV2-CM or CXCL5 (Figure 2E-2G). Although we did not specifically examine whether the inhibitor affects Akt activation, our results provide sufficient evidence to establish that CXCL5 is a crucial factor in promoting VSMC migration. Moreover, we are interested in conducting further investigations to study the specific signaling pathways associated with the CXCL5/CXCL2 axis in VSMC migration, aiming to gain a more detailed understanding of this process.
Reviewer 2 Report
The work of Sun and colleagues focuses on the abnormal communication between EC and VSMC during atherosclerosis. They specifically investigate the changes exerted by ETV2 during EC injury and remodeling. The researchers demonstrate that ETV2-overexpressed ECs release cytokines, with CXCL5 playing a pivotal role in VSMC migration through Akt, p38, and c-jun-dependent mechanisms. The study is supported by in vitro experiments, starting from ETV2-overexpressed EC conditioned medium to CXCL5 activation. While the work is of interest, there are several issues that should be addressed before publication:
1 -In the abstract, lane 26, the use of the conjunction "however" is misleading.
2- In the introduction, please expand on the sentence in lane 40-41 regarding the interaction between ECs and VSMCs.
3- Does ad-ETV2 CM affect CXCR2 expression on VSMCs?
4- Since ad-ETV2 CM contains different cytokines, please provide an experiment in which a blocking antibody against CXCL5 is added, as stated in the discussion (lane 270-271).
5 - Figure legend 1, lane 101, please change the scale bar to 100 µm instead of 100 µM.
6- Figure legend 1, lane 104, the lines of the wound are not broken.
7- Results 2.2: The sentence in lane 124 is misleading because the entire study does not solely focus on CXCL5.
8 - Results 2.3: Do the expression levels of the mentioned genes also change with the conditioned medium (CM)?
9- Could you provide real-time or protein expression data for ETV2 in HUVECs?
10- the discussion section needs to provide a deeper understanding of the role of ETV2 in vessel injury and establish a stronger link with atherosclerosis.
11- Methods 4.6 and 4.7 refer to VSMCs, not HUVECs.
12- Please rearrange the figures for better clarity and understanding. Place the relevant graph of images below or to the right consistently.
Author Response
We thank the reviewer for the positive comments and detailed review: “The work of Sun and colleagues focuses on the abnormal communication between EC and VSMC during atherosclerosis. They specifically investigate the changes exerted by ETV2 during EC injury and remodeling. The researchers demonstrate that ETV2-overexpressed ECs release cytokines, with CXCL5 playing a pivotal role in VSMC migration through Akt, p38, and c-jun-dependent mechanisms. The study is supported by in vitro experiments, starting from ETV2-overexpressed EC conditioned medium to CXCL5 activation. While the work is of interest, there are several issues that should be addressed before publication.”
Our responses to the reviewer’s suggestions/comments are detailed below:
- In the abstract, lane 26, the use of the conjunction "however" is misleading.
As suggested, we have replaced the term "however" with "in addition" to better convey the intended meaning.
- In the introduction, please expand on the sentence in lane 40-41 regarding the interaction between ECs and VSMCs.
As suggested by the reviewer, we have included additional explanations regarding the interaction between endothelial cells (ECs) and vascular smooth muscle cells (VSMCs). The following statement has been incorporated into our manuscript: “The vasculature plays a pivotal role in supplying blood and oxygen to all body tissues within the body, making it significant in the treatment of conditions involving ischemia and injury-induced regeneration [7]. Endothelial cells have the ability to secrete vasoactive agents and reactive oxygen species (ROS), which can influence the function of vascular smooth muscle cells and subsequently regulate vascular function [8].”
- Does ad-ETV2 CM affect CXCR2 expression on VSMCs?
In our study, our primary objective was to investigate the direct effects of altered cytokines secreted by HUVECs due to ETV2 on VSMC migration. As such, we did not specifically assess whether CXCR2 expression was affected. However, we acknowledge that examining the impact of ETV2 on CXCR2 expression could be an interesting issue for future research, as it may provide further insights into the underlying mechanisms involved in VSMC migration.
- Since ad-ETV2 CM contains different cytokines, please provide an experiment in which a blocking antibody against CXCL5 is added, as stated in the discussion (lane 270-271).
As suggested by the reviewer, we have incorporated the results of the antibody blocking experiment in Figure 2F. Consequently, we have modified the Figure legend for clarity.
- Figure legend 1, lane 101, please change the scale bar to 100 µm instead of 100 µM.
As suggested by the reviewer, we have modified the unit correctly.
- Figure legend 1, lane 104, the lines of the wound are not broken.
As suggested by the reviewer, we have deleted the word “broken”.
- Results 2.2: The sentence in lane 124 is misleading because the entire study does not solely focus on CXCL5.
We appreciate the reviewer's comment, and we have made the following modification to the sentence: "Given that CXCL5 exhibited the highest alteration among the various cytokines secreted by ETV2-transduced HUVECs, we focused on investigating the effects of CXCL5 on VSMC migration."
- Results 2.3: Do the expression levels of the mentioned genes also change with the conditioned medium (CM)?
Regrettably, we did not evaluate the alteration of MMPs in VSMCs by the conditioned media. However, we appreciate the reviewer's suggestion and we intend to conduct a more comprehensive investigation into the role of ETV2 in vascular physiology in future studies.
- Could you provide real-time or protein expression data for ETV2 in HUVECs?
As suggested by the reviewer, we demonstrated successful expression of ETV2 in HUVECs through western blot and immunocytochemistry, which were added in Figs 1A and 1B. However, we would like to mention that we were unable to generate an expression graph for the western blot due to the absence of detectable endogenous ETV2.
- the discussion section needs to provide a deeper understanding of the role of ETV2 in vessel injury and establish a stronger link with atherosclerosis.
We appreciate the reviewer's comment, and we have made the following modification to the sentence: “In addition, ETV2 has been reported as a potential therapeutic target for myocardial infarction and a key regulator of vascular regeneration in hind limb animal models through endothelial remodeling. Following injury, the expression of ETV2 was readily detected in endothelial cells. Subsequently, there was a notable increase in vessel formation and neovascularization induced by Etv2. Because vascular injury plays a crucial role as an initial trigger in the development of different vascular disorders, such as atherosclerosis [48], ETV2 may serve as an important factor linking endothelial cells and smooth muscle cells in response to vascular injury."
- Methods 4.6 and 4.7 refer to VSMCs, not HUVECs.
As suggested by the reviewer, we have corrected the cell name.
- Please rearrange the figures for better clarity and understanding. Place the relevant graph of images below or to the right consistently.
As suggested by the reviewer, we have placed the relevant graph of images to the right consistently except for Fig. 5C.